# Robotic Retrograde Intrarenal Surgery: A Journey from “Back to the Future”

**DOI:** 10.3390/jcm11185488

**Published:** 2022-09-19

**Authors:** Vineet Gauhar, Olivier Traxer, Sung-Yong Cho, Jeremy Yuen-Chun Teoh, Alba Sierra, Vishesh Gauhar, Kemal Sarica, Bhaskar Somani, Daniele Castellani

**Affiliations:** 1Department of Urology, Ng Teng Fong General Hospital (NUHS), Singapore 609606, Singapore; 2Department of Urology AP-HP, Tenon Hospital, Sorbonne University, F-75020 Paris, France; 3Department of Urology, Seoul National University Hospital, Seoul 02841, Korea; 4S.H.Ho Urology Centre, Department of Surgery, The Chinese University of Hong Kong, Hong Kong 999077, China; 5Global Indian International School, Singapore 828649, Singapore; 6Department of Urology, Biruni University Medical School, Istanbul 34010, Turkey; 7Department of Urology, University Hospitals Southampton, NHS Trust, Southampton SO16 6YD, UK; 8Urology Unit, Azienda Ospedaliero-Universitaria Ospedali Riuniti di Ancona, Università Politecnica delle Marche, 60126 Ancona, Italy

**Keywords:** renal stones, robotics, retrograde intrarenal surgery, flexible ureteroscopy

## Abstract

The introduction of robotics has revolutionized surgery. Robotic platforms have also recently been introduced in clinical practice specifically for flexible ureteroscopy. In this paper, we look at the robotic platforms currently available for flexible ureteroscopy, describing their advantages and limitations. The following robotic platforms are discussed: Roboflex Avicenna^®^, EasyUretero^®^, and ILY^®^ robot. Finally, potential future advancements in this field are presented.

## 1. Introduction: Robotics and Urology

A new milestone in the history of surgery was achieved when robotic assistance in laparoscopic surgery was introduced at the turn of this millennium. First introduced in 1999, the Da Vinci surgical system (Intuitive Surgical, Sunnyvale, CA, USA), was approved for human use by the FDA in 2000. Urologists were early adopters of robotic technology and 7000 da Vinci surgical systems were installed in 69 countries [1]. Currently, five other robotic platforms are commercially available, expanding the use of robotic-assisted surgery [2].

One of the first applications of a novel flexible robotic technology in endourology, which was originally devised to assist electrophysiological ablation for cardiac arrhythmia [3] (i.e., novel flexible Sensei^®^ robotic catheter system), was reported by Desai et al. [4]. In this study, 18 patients with renal calculi underwent flexible robotic ureteroscopy wherein after manually introducing the device into the renal collecting system over a guidewire under fluoroscopic control, a custom-built 7.4 Fr flexible ureteroscope was passed through the device. After this, all intrarenal manoeuvers, including stone relocation and fragmentation into 1 to 2 mm particles, were done exclusively from the remote robotic console. This human study was then considered a path-breaking procedure, but no further studies were ever reported. The authors mentioned that the robotic system had several limitations. The used diameter of the robotic catheter system was too large. It needed to be decreased to less than 14 Fr, especially if routine ureteral pre-stenting is to be avoided. The construct of the system did not allow full robotic manipulation from the lower to the upper tract and robotic manipulation was limited only to the area within the renal collecting system.

Most of the aforementioned systems are used in non-endourological interventions; there is to date only one robot designed specifically for flexible ureteroscopy (F-URS). Since 2012, ELMED (Ankara, Turkey) has been working on a robot specifically designed for F-URS called the Roboflex Avicenna (RA) [5]. The IDEAL (idea, development, evaluation, assessment, long-term study) framework for stages in surgical innovation was introduced, specifically focused on robotic surgery, and the RA underwent clinical testing as per protocols of IDEAL stage 2 recommendations by experienced endourologists [6].

## 2. Robotic Ureteroscopy System: Back to Roboflex Avicenna

The design system of the RA consists of two elements: the surgeon’s console with a chair and the manipulator of the flexible ureterorenoscope (Figure 1). During the developmental phase, several significant improvements were made to the size and design of the function screen; the design of the joysticks, to control rotation and deflection of the endoscope; the fine adjustment of deflection of the endoscope by the central wheel; and the range of rotation of the manipulator.

The console allows for a surgeon to be seated and control the manipulator via two joysticks that allow up and down deflection and side-to-side manipulation. The control monitor with its touch screen access manipulates the advancement and retraction of the laser fiber and adjustment of the irrigation flow rate. In addition, the degree of rotation and deflection is displayed. Two foot pedals are integrated to control the laser device and fluoroscopy. The manipulator consists of the motor system and the robotic arm, which holds and moves the endoscope at an adjustable height to match the patient’s anatomy. Any endoscope can be fixed in the arm and further stabilized by two holders during manipulation and the manipulator directly drives the flexible ureteroscope using its own mechanics. Several functions could be also integrated, such as fine-tuning of the movements, motorized insertion and retraction of the laser fiber, and automatic repositioning for the introduction of the fiber. The entire assembly was aimed at improving functionality while performing RIRS, with ergonomic ease as its foundation.

In the first reported clinical study by Saglam et al., 81 patients (mean age: 42 years, range: 6–68) were treated in an IDEAL stage 2 study by seven experienced urologists with 5–16 years of experience in F-URS using the RA [5]. They were able to objectively infer that the robotic platform significantly improved ergonomics (total questionnaire score: 5.6 vs. 31.3). After only a short introduction to the device in the training model, all seven surgeons were able to perform robot-assisted F-URS safely and in a reasonable time frame compared with their own published series of classic F-URS [7,8]. Other benefits attributed were lesser possibility of scope damage because the functions of the RA, such as insertion of the laser fiber only in a straight position of the scope using a memory function, stepwise motorized advancement of the laser fiber, and force-controlled (maximal 1 N/mm^2^) deflection of the scope. All those benefits helped mitigate human error as well as allowing more precision and mitigating fatigue during laser lithotripsy. These could improve efficiency and efficacy of the F-URS procedure, allowing even larger stones to be tackled with good results.

Klein et al. reported a prospective case series of 395 patients undergoing robotic F-URS with the Avicenna system [9,10]. They demonstrated the system to be safe and easy to integrate with daily routine. Like Geavlete et al. [11], they also observed a subjectively better operation comfort for robotic F-URS. Proietti et al. in a ureteroscopy simulation test using the RA were able to show that spatial orientation and scope stability were quite good with the RA even in subjects with no prior ureteroscopy training [12]. Sarica et al. showed that robotic F-URS using RA was helpful during combination treatment of multicalyceal stones when combined with supine mini-percutaneous nephrolithotomy (PCNL), and reaffirmed previous findings that it saves time and protects the laser fibers and prevents fatigue of the user [13].

As he shows in Figure 2, the ergonomics of the console and a near total out of radiation zone during fluoroscopy for the surgeon are unparalleled dual advantages.

Limitations of the robotic system are a lack of tactile feedback and problems with the use of baskets for extraction of larger stone fragments, yet as RA enables graduated movements of the scope in all three dimensions, it overcomes the lack of tactile feedback. Robotic F-URS requires partial undocking of the device, which may be cumbersome and time-consuming. Finally, the costs of the device may become a significant issue, particularly concerning the financial restrictions of health care systems. Recently during the COVID-19 outbreak, Al-Ansar et al. reported that they found managing COVID-19-positive patients with robotic F-URS helped minimize the direct contact with the patient and speed up the procedure [14], which is recommended as a good urology action plan in a pandemic [15]. 

## 3. Robotic Ureteroscopy: Insight into the Future

Renewed interest in surgical robots has recently been cited by Peters et al., as the robot system can accurately and reliably control the surgery with good task repeatability [16]. When we consider the advantages of a robotic F-URS system, the reduction of work-related musculoskeletal fatigue and the risk of radiation exposure become its most significant contributions.

### 3.1. The EasyUretero^®^


A new robotic flexible ureteroscopic system made in Korea has been recently introduced, the EasyUretero^®^ (Roen Surgical, Inc., Daejeon, Korea). The system is a master-slave robotic system like the Da Vinci robotic system. It consists of the master’s console and the arm in the slave robot for manipulating the flexible ureteroscopes. The current model of the master’s console has a 32-inch touch screen, an armrest, a gimbal handle, and a clutch pedal for handling the flexible ureteroscopes. The monitor can display the inserted part of the instrument by transmitting the view of the flexible ureteroscopes. The prototype handle looks like a television remote controller with a jog wheel for movement of the laser fiber and the stone baskets without assistance. Some buttons in this handle are for precise movement of the flexible ureteroscopes. The handle can make forward and backward movements, upward and downward deflections, and rotate the flexible ureteroscopes (Figure 3).

The current version of the ureteroscope mounting part in the slave robot can mount some commercially available flexible ureteroscopes, such as LithoVue (Boston Scientific, Marlborough, MA, USA) and Flex-Xc (Karl Storz Endoskope, Tuttlingen, Germany). Uniquely, this robotic system has an automated navigation system from the ureteral access sheath (UAS) to the targeted stones and a flexible ureteroscopic movement can be recorded and replayed. The function can be helpful in repetitive tasks such as the retrieval of multiple fragmented stones using a stone basket (Appendix A). Additionally, it has a safety function to measure and mark the oversized fragmented stones compared to the diameter of the end of the UAS. Its ergonomic design, open console, and seated concept make this robot an ergonomic delight for urologists and even repetitive actions such as basket retrieval of fragments are much easier to perform with an easy-to-use interface on the touch console (Figure 4).

A recent animal study by Han et al. in 2022 showed the feasibility of the Easy Uretero robotic platform by retrieving the renal stones in four female pigs [17]. The 1.0–1.5 cm calcium oxalate stones were inserted through the nephrostomy tract and an 18 Fr Amplatz renal sheath formed under ultrasound guidance. Three urological surgeons experienced in a variable number of flexible ureteroscopic cases participated in the study. All stones were successfully removed, and it was concluded that the easy Uretero robotic flexible ureteroscopic system was feasible and safe even for less-experienced surgeons.

The human clinical trial of the EasyUretero^®^ system was successfully completed in April 2022 at Seoul National University Hospital and Severance Hospital, Yonsei University College of Medicine, Seoul, Korea. It was a multicentered, prospective, single-arm, and pivotal study, and enrolled 47 patients. The final results will be reported shortly (Registry: WHO International Clinical Trials Registry Platform, Clinical Research Information Service [http://cris.nih.go.kr]; Identifier: KCT0007506). The current version of the ureteroscope mounting part in the slave robot can mount only a limited number of flexible ureteroscopes. Hopefully, the next version of the robotic platform can adopt any kind of flexible ureteroscope in the ureteroscope mounting part according to the manufacturers’ details. Additionally, the next version would ideally embed some automated functions such as an automated navigation system in the renal calyces, respiration-adjusted stone fragmentation function, measurement of some operative parameters of laser setting, irrigation sets, intrarenal pressure, and temperature according to the manufacturers’ blueprint.

### 3.2. ILY^®^ Robot: A Ureteroscopic Holder with Multiple Degrees of Freedom

ILY^®^ (STERLAB, Vallauris, France) is a ureteroscope holder with multiple degrees of freedom telemanipulated with a wireless controller (Figure 5). The ILY^®^ robot is placed close to the patient and remotely controlled by the surgeon thanks to a wireless console for F-URS.

Unlike the rest of the robots destined for endourology, ILY^®^ uses a remote control, like PlayStation’s joystick (Figure 6), which allows the surgeon to work easily at a distance from the patient and keeps them away from the source of ionizing radiation.

The ILY^®^ ureteroscope holder is compatible with all types of ureteroscopes (reusable and single-use) (Figure 7). To date, it is compatible with all the flexible ureterorenoscopes (produced by the following companies: Wolf, Storz, Olympus, Pusen, Boston Scientific, Seplou, Hugemed, Innovex, Seegen, Hawk, Redpine, and Scivita).

With the ILY^®^ robot, it is also possible to fix the UAS. This ureteroscope holder is compatible with most of the brands of sheaths available in the market and in different sizes. One can select in the tactile screen, during UAS placement, the type and size of the sheath and the sheath holder modulates according to it.

#### Using the ILY^®^ Robot

The patient is placed in the lithotomy position under general anesthesia. As a first step, we start placing a guide wire into the renal pelvis; then, we introduce a UAS through. When the UAS is placed in its final position, close to the ureteropelvic junction, we fix the UAS to the ILY^®^ robot holder. As mentioned before, this receptacle is compatible with all UAS types and sizes. We move the robot adjacent to the patient and we select the size and the brand of the UAS that we are using to fix it in the robot. Then, through the UAS, we introduce the flexible ureteroscope to reach the renal pelvis, and once we arrive at the pelvis, we fix the flexible ureteroscope. When both UAS and ureteroscope are properly fixed, we activate the remote control to move the scope inside the renal pelvis, to reach the stone. You can go back and forth, with the left and right frontal bottom, respectively. With the joystick, you can move the distal part of the ureteroscope, and you can flex and deflect and twerk left and right, with the right and the left joystick, respectively (Figure 6). Once the flexible ureteroscope is placed into the kidney, the collecting system is explored, starting with the upper calyces, followed by the middle and lower calyces. In the right kidney, the calyces are seen on the left of the endoscope screen. Thus, supination is the essential movement to explore the right renal cavities using the left joystick. Conversely, the left renal calyces are seen on the right of the endoscope screen, meaning that pronation is essential when exploring the left kidney, so you need to move the left joystick to the right. Once the stone is placed in front of the flexible ureteroscope, we can proceed to laser lithotripsy and small movements are needed with the remote control (Figure 8). The major advantage regarding maneuverability is its increased ability to rotate (+/−360°). In addition, as we place the scope over a surface, we gain stability and accuracy, reducing the movements of the scope and fiber tip during the renal stone’s lithotripsy.

The importance of ergonomics in F-URS, as mentioned by Ong et al. and Gabrielson et al. [18,19], is essential for surgeons who perform several surgeries per day and longer procedures. The wireless console relieves the surgeon from the tedious manual handling of the ureteroscope, which causes fatigue and osteoarticular pain in the upper limbs, and in addition the surgeon can be placed seated during the procedure and perform small movements with the remote control. Placing the robot is easy and quick, which avoids extra operative time. In our experience, it takes only 5 min to set up the robot and 3 min to install it. As a limitation, while other robots try to reproduce hand movements such as the ones performed during traditional F-URS, the ILY^®^ robot is not very intuitive. Using a video-game controller forces the surgeon to learn how to use ILY^®^ control buttons and initially it is time-consuming, because a surgeon needs to familiarize themself with the control buttons to replicate the flexible ureteroscope movement that is expected. However, once mastered, we could reproduce all the movements with multiple degrees like in standard F-URS.

## 4. Potential Future Advancements

The future of robotic F-URS could be seen in another prototype proposal for a novel master-slave controlled robotic system for F-URS from China, which integrates the function of intra-renal pressure monitoring and an optimization method based on neural networks to make the operation of the above haptic device more convenient and intelligent [20]. The authors propose that their study also opens the pathway for an intelligent sensing algorithm for robotic F-URS as a potential research direction in the future.

Among emerging technologies, the new multispeciality Monarch^®^ platform (Auris Health, Redwood City, CA, USA) for robotic ureteroscopy (Auris Health, Redwood City, CA, USA) has been recently approved for urological procedures in the United States [21]. The Monarch platform allows for robotic control of the ureteroscope with an X-Box-type controller with excellent visualization. Benefits include stability, ease of navigation, ergonomics, and reduced radiation exposure to the surgeon (https://www.aurishealth.com/monarch-platform; (accessed on 9 September 2022). Additionally, new software can allow for automated actions such as sized detection, laser settings, control of respiration, electromagnetic targeting, and artificial intelligence applications. In addition, the Monarch^®^ platform supports percutaneous nephrolithotomy procedures, too. It reduces the complexity of gaining percutaneous access and aids stone clearance efficiency through simultaneous fragmentation and suctioning of stones with robotic assistance [21].

Fifth-generation technology and telesurgery are making a remarkable breakthrough even in urology as seen by the successful completion of 29 robotic controlled remotely operated radical nephrectomies in 8 primary hospitals in China [22]. Robotic F-URS may be the start of a new lease on life with future implements enabling surgeons to perform telerobotic F-URS by manipulating the console, probably just using their handphone [23]. 

Adapting and learning from the excellence that the Da-Vinci platform has achieved with specific instruments designed for the robot based on surgeons’ needs and utility, it may be essential that in future robotic F-URS offers instruments designed to improve the robot’s performance as much as ergonomics.

Lastly, for the future of robotic surgery, miniaturization is a key. In the future, integrating camera robots and microrobots should allow a mobile viewing platform and minimize the operative theater carbon footprint [24]. That is where absolute flexibility will meet precision functionality, and the future is already here as shown by Micromate, the first FDA-approved miniature surgical robot for needle procedures [25].

## 5. Conclusions

The development of new robotic systems has expanded the potential advantages of robotic uretero-renoscopic surgery by enabling surgeons to perform the procedure with less radiation exposure, and improved ergonomic conditions, particularly in cases with large/complex and multiple stones. Robotic RIRS will be suitable for all but especially for large and complex stones, which require precision and longer lithotripsy time. This allows for ergonomical and controlled RIRS. With less fatigue, robotic RIRS allows for more time and ability for better inspection of all calyces, which can ensure lower residual fragments. The technical and technological development of these easy-to-use master-slave systems continue to add merit to this surgery, which seems to have a certain role in near future endourological stone management. However, this application may currently be limited only to centers where the robot is accessible and affordable.

## Figures and Tables

**Figure 1 jcm-11-05488-f001:**
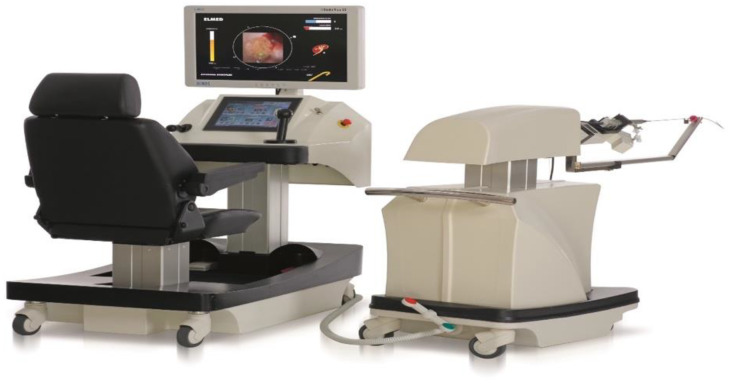
Roboflex Avicenna: console with chair and manipulator for scope.

**Figure 2 jcm-11-05488-f002:**
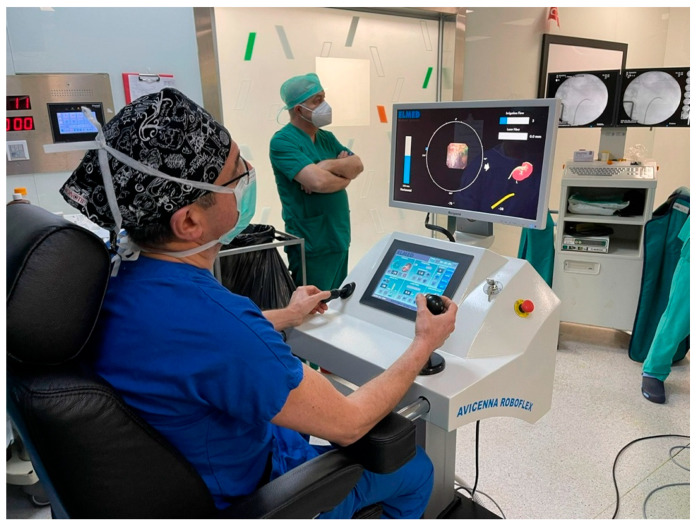
Surgeon working at the Roboflex Avicenna console.

**Figure 3 jcm-11-05488-f003:**
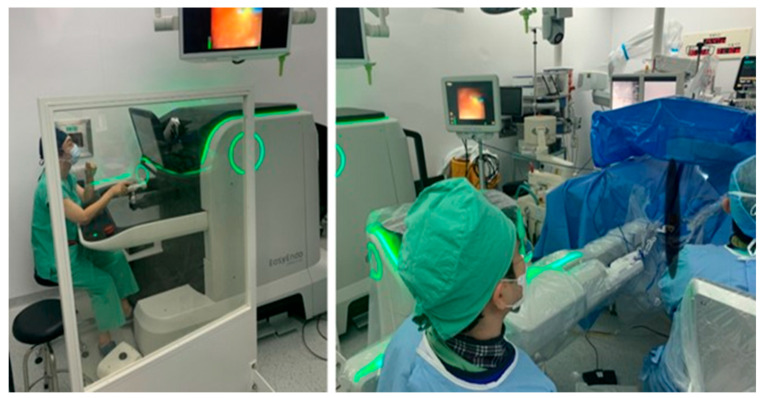
General view of the prototype of the EasyUretero^®^ robotic flexible ureteroscopic system: master’s console on the **left**; the slave robot on the **right**.

**Figure 4 jcm-11-05488-f004:**
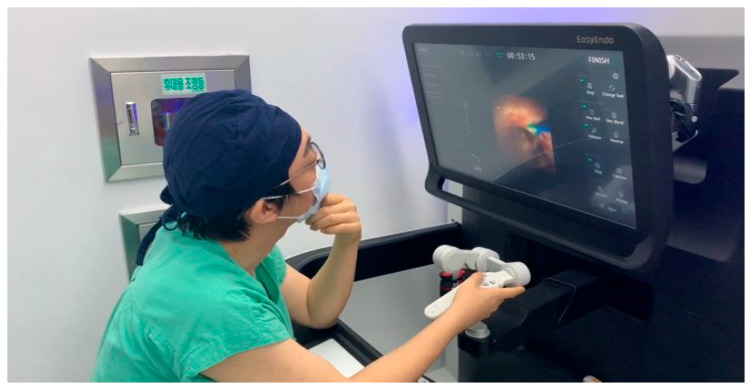
Easy Uretero^®^ robotic console navigation and basket manipulation function.

**Figure 5 jcm-11-05488-f005:**
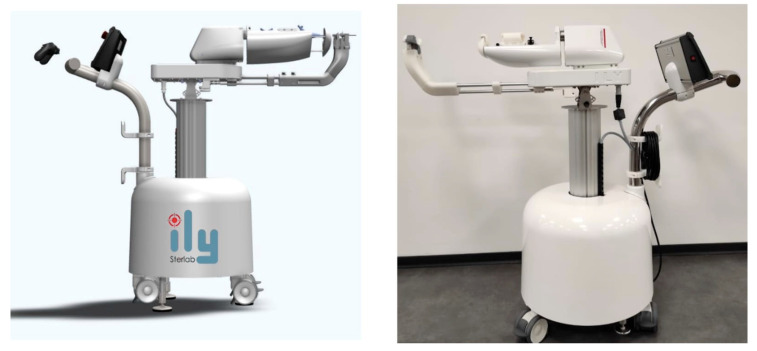
ILY^®^ robot.

**Figure 6 jcm-11-05488-f006:**
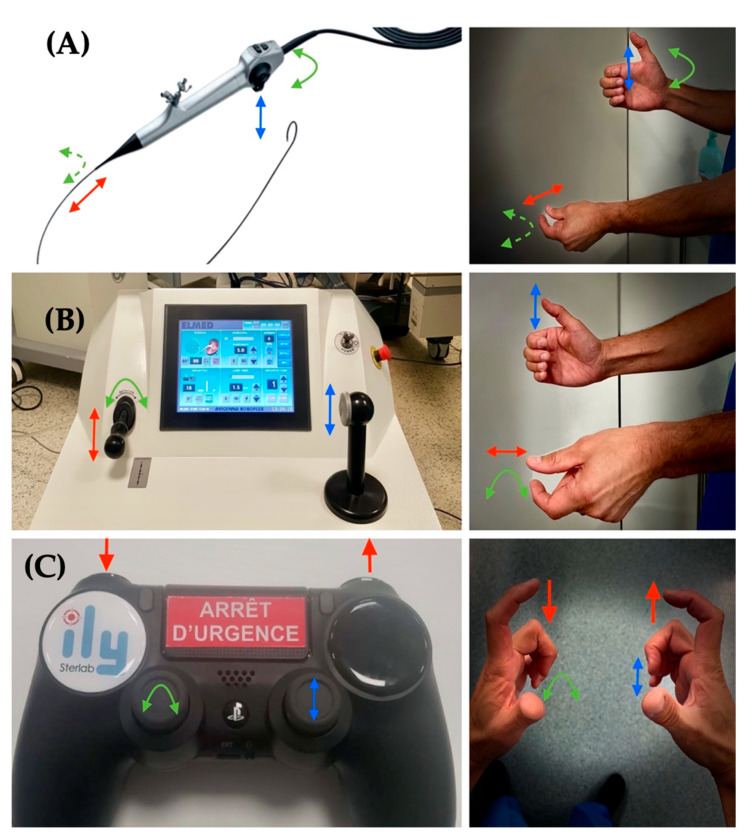
Placement of surgeon’s hands and movements to manipulate (**A**): Flexible ureterorenoscope; (**B**): Avicenna robot; (**C**): ILY^®^ robot. Arrows represent how to perform different movements. Avicenna tries to reproduce hand movements, while ILY^®^ changes completely the concept of using a video-game controller. Arrows indicate directions of hands, fingers, scope and movements.

**Figure 7 jcm-11-05488-f007:**
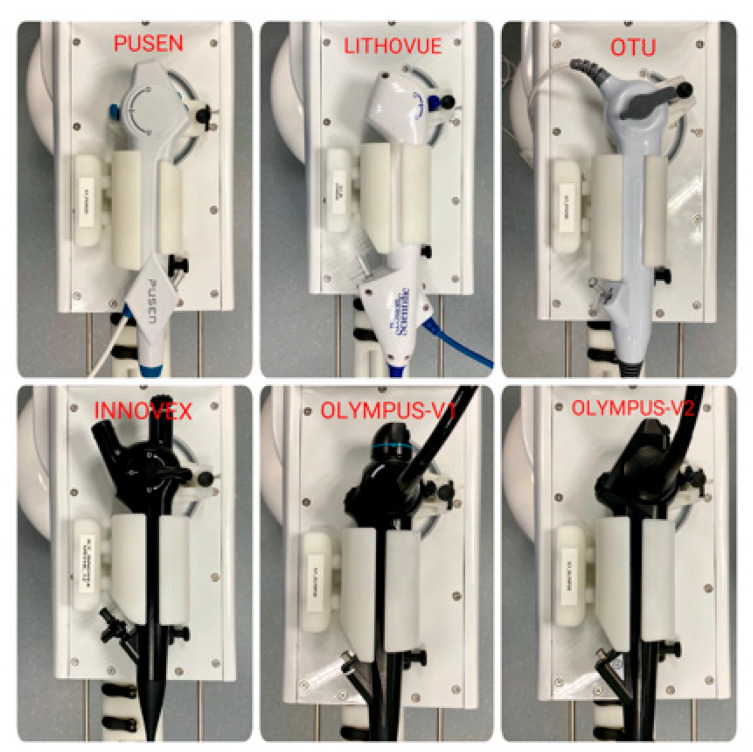
The ILY^®^ ureteroscope holder is compatible with all types of ureteroscopes available on the market (reusable and single-use).

**Figure 8 jcm-11-05488-f008:**
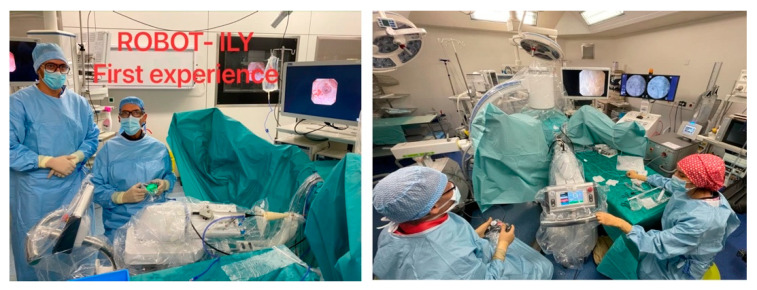
Surgeon handling the remote control for laser lithotripsy.

## Data Availability

Not applicable.

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
