# Peer review of "Robotic Retrograde Intrarenal Surgery: A Journey from “Back to the Future”"

_jcm, 2022, doi:10.3390/jcm11185488_

Round 1
Reviewer 1 Report
This is a review paper discussing the different robotic system currently available or under development for urological applications. It discusses the designs of each platform with the clinical experience with each system. The paper is well written.
I have only one comment for the authors. Can they provide some information about the clinical cases in which these robotic systems would be most useful? I’m sure they can be used for all cases, but are there any specific cases that would benefit greatly from the robotic systems?
Author Response
This is a review paper discussing the different robotic system currently available or under development for urological applications. It discusses the designs of each platform with the clinical experience with each system. The paper is well written. I have only one comment for the authors. Can they provide some information about the clinical cases in which these robotic systems would be most useful? I’m sure they can be used for all cases, but are there any specific cases that would benefit greatly from the robotic systems?
REPLY.
We would like to thank the reviewer for this nice comment on our study. We have added the following sentence in Conclusion of the revised manuscript “Robotic RIRS will be suitable for all but especially for large and complex stones which require precision and longer lithotripsy time. This allows for ergonomically and controlled RIRS. With lesser fatigue, robotic RIRS allows for more time and ability for better inspection of all calyces which can ensure lower residual fragments.” Thank you very much for suggesting this.
Reviewer 2 Report
1) General comments
This is a review article regarding with robotic RIRS system over the world in current era. Robotic RIRS is ongoing inventive topics in urological field. The endourological procedures using fURS into renal collecting system is one of hard procedures for surgeons because of difficulty of manipulation, wearing the x-ray protector, and radiation exposure during RIRS. Therefore, robotic surgical system will be required to mitigate and overcome this hardship. This review article includes many novel topics and current state regarding world endourological robotics. It is very valuable. I want to request the following.
2) Comments for revisions
①I am interested in the Monarch system. Can you use some pictures&photo in this review? Many readers are going to hope to see that system in this article.
Author Response
1) General comments
This is a review article regarding with robotic RIRS system over the world in current era. Robotic RIRS is ongoing inventive topics in urological field. The endourological procedures using fURS into renal collecting system is one of hard procedures for surgeons because of difficulty of manipulation, wearing the x-ray protector, and radiation exposure during RIRS. Therefore, robotic surgical system will be required to mitigate and overcome this hardship. This review article includes many novel topics and current state regarding world endourological robotics. It is very valuable. I want to request the following.
REPLY. We would like to thank the reviewer for this nice comment on our study
2) Comments for revisions
I am interested in the Monarch system. Can you use some pictures&photo in this review? Many readers are going to hope to see that system in this article.
REPLY.
We would like to thank the reviewer for suggesting us this. Unfortunately, monarch has no pictures to provide without copyright as none of the authors are directly associated with it. We added the webpage link in the revised version of the manuscript.